

# Genetic diversity and SNP's from the chloroplast coding regions of virus-infected cassava

Bruno Rossitto De Marchi[1,3], Tonny Kinene[2], Renate Krause-Sakate[3], Laura M. Boykin[2], Joseph Ndunguru[4], Monica Kehoe[5], Elijah Ateka[6], Fred Tairo[4], Jamisse Amisse[7] and Peter Sseruwagi[4]

[1] Gulf Coast Research and Education Center, University of Florida, Wimauma, FL, USA
[2] School of Molecular Sciences and Australian Research Council Centre of Excellence in Plant Energy Biology, University of Western Australia, Crawley, Perth, WA, Australia
[3] Department of Plant Protection, UNESP - Universidade Estadual Paulista Julio de Mesquita Filho, FCA, Botucatu, São Paulo, Brazil
[4] Mikocheni Agricultural Research Institute, Dar es Salaam, Tanzania
[5] Department of Primary Industries and Regional Development Diagnostic Laboratory Service, South Perth, WA, Australia
[6] Department of Horticulture, Jomo Kenyatta University of Agriculture and Technology, Nairobi, Kenya
[7] Mozambique Agricultural Research Institute, Nampula, Mozambique

Corresponding authors
Bruno Rossitto De Marchi,
bruno.rossittode@ufl.edu
Laura M. Boykin, lboykin@mac.com

## ABSTRACT

Cassava is a staple food crop in sub-Saharan Africa; it is a rich source of carbohydrates and proteins which currently supports livelihoods of more than 800 million people worldwide. However, its continued production is at stake due to vector-transmitted diseases such as Cassava mosaic disease and Cassava brown streak disease. Currently, the management and control of viral diseases in cassava relies mainly on virus-resistant cultivars of cassava. Thus, the discovery of new target genes for plant virus resistance is essential for the development of more cassava varieties by conventional breeding or genetic engineering. The chloroplast is a common target for plant viruses propagation and is also a potential source for discovering new resistant genes for plant breeding. Non-infected and infected cassava leaf samples were obtained from different locations of East Africa in Tanzania, Kenya and Mozambique. RNA extraction followed by cDNA library preparation and Illumina sequencing was performed. Assembling and mapping of the reads were carried out and 33 partial chloroplast genomes were obtained. Bayesian phylogenetic analysis from 55 chloroplast protein-coding genes of a dataset with 39 taxa was performed and the single nucleotide polymorphisms for the chloroplast dataset were identified. Phylogenetic analysis revealed considerable genetic diversity present in chloroplast partial genome among cultivated cassava of East Africa. The results obtained may supplement data of previously selected resistant materials and aid breeding programs to find diversity and achieve resistance for new cassava varieties.

## INTRODUCTION

Cassava, *Manihot esculenta* Crantz subsp. *esculenta* (Euphorbiaceae) is a perennial woody shrub also known as manioc or tapioca. Cassava is globally cultivated for its starchy storage roots, providing staple food for over 800 million people throughout Africa, Asia and South America (http://faostat.fao.org). Africa is currently the biggest cassava producer in the world (*Hillocks & Thresh, 2002*). Cassava is highly resistant to drought and its roots can be maintained in the soil for a couple of years until harvesting, which makes this crop an important reserve of carbohydrates to relieve global famine (*Raheem & Chukwuma, 2001*). The cassava propagation process from season to season is mostly done by the use of stem cuttings. The vegetative propagation exposes the crop to the proliferation of several pathogens, including virus diseases such as Cassava mosaic disease (CMD) and Cassava brown streak disease (CBSD). These viral diseases are threating cassava's continued production in all major growing areas of Africa. The production losses caused by CMD and CBSD together in Africa were reported to be more than US $3 billion every year (*Hillocks & Jennings, 2003*; *Legg et al., 2006*). CMD is caused by several different geminivirus species including *African cassava mosaic virus*, *East African cassava mosaic virus*, and *East African cassava mosaic virus*—(Uganda) (*Legg et al., 2006*). CBSD was primarily thought to be caused by two different Ipomovirus species, the *Cassava brown streak virus* (CBSV) and the *Ugandan cassava brown streak virus* (UCBSV). However, more recent studies have found that up to four different CBSVs species might be associated with CBSD (*Ndunguru et al., 2015*). CBSD incidences are higher near to the coast of Tanzania and Mozambique (*Hillocks & Jennings, 2003*). These viruses are transmitted by the whitefly, *Bemisia tabaci* (Gennadius) which is one of the most serious pests of agriculture worldwide (*Lowe et al., 2000*). In East Africa, high populations of whiteflies are visible in cassava crops and are causing tremendous yield losses (*Hillocks & Jennings, 2003*). However, a recent study suggests that aphids may also play a role in the transmission of CBSV, due to the occurrence of a conserved DAG motif that is associated with the aphid vector (*Ateka et al., 2017*). Currently, the management and control of viral diseases in cassava relies mainly on tolerant cultivars. The main goal of the breeders is to obtain cultivars tolerant to both diseases. Although cassava with high tolerance for CMD has been reported, the sources of resistance to CBSD are very restricted (*Tumwegamire et al., 2018*). Thus, new target genes for plant virus resistance are essential for the development of virus-resistant cassava. The chloroplast components are a common target for plant virus propagation and can also play a role in plant defense against viruses (*De Torres Zabala et al., 2015*). One of the defense lines developed by plants against pathogens is based on plant resistance proteins which detect pathogens effectors activating an effector-triggered immunity (ETI) system (*Dangl, Horvath & Staskawicz, 2013*; *Stuart, Paquette & Boyer, 2013*). A very well characterized ETI is the hypersensitive response, a localized form of programed cell death in the surrounding area of the infection site that relies on a burst of reactive oxygen species in chloroplasts (*Zurbriggen, Carrillo & Hajirezaei, 2010*). In addition, the chloroplast is associated with the synthesis of key plant defense mediators, such as the hormones salicylic acid, abscisic acid and jasmonic

acid (*Serrano, Audran & Rivas, 2016*; *Nambara & Marion-Poll, 2005*). There are several chloroplast factors reported to be interacting with virus nucleic acids or proteins (*Zhao et al., 2016*). Therefore, this organelle is a potential source for discovering new resistant genes for plant breeding. A series of typical changes followed by chlorotic symptoms imply the occurrence of chloroplast–virus interactions (*Zhao et al., 2016*; *Dardick, 2007*; *Mochizuki et al., 2014*). Many plant viruses have as a signature infection pattern the attachment to the chloroplast membrane, which affects large numbers of chloroplast and photosynthesis-related genes (*Wei et al., 2010*; *Bhattacharyya & Chakraborty, 2018*). In other words, damage to the chloroplast is a fundamental step in successful infection for plant viruses. Certainly, several chloroplast factors have been identified to interact with viral components that could lead a virus-tolerance to the host plant (*Qiao et al., 2009*; *Lim et al., 2010*). These factors are involved in virus replication, movement, symptoms or plant defense, suggesting that viruses have evolved to interact with chloroplast (*Zhao et al., 2016*). Currently, more than 800 complete chloroplast genome sequences are available in the National Center for Biotechnology Information organelle genome database, including the complete cassava chloroplast genome.

The cassava chloroplast genome is 161,453 bp in length and includes a pair of inverted repeats (IR) of 26,954 bp. It consists of 128 genes; 96 are single copy and 16 are duplicated in the IR. There are four rRNA genes and 30 distinct tRNAs, seven of which are duplicated in the IR (*Daniell et al., 2008*). Complete chloroplast genome sequences, as well as other regions of the chloroplast genome, have been proved to be useful tools for improving our understanding of the phylogenetic relationships between closely related taxa and the evolution of plant species (*Daniell et al., 2016*; *Xu et al., 2017*). In addition, they are used for the accurate identification of commercial cultivars and determination of their purity (*Daniell et al., 2016*). The goal of this study was to evaluate the genetic diversity of cassava chloroplast coding regions from plants naturally infected with viruses from different varieties collected in the field in East Africa. Other sequences from GenBank, including cassava wild relatives, were added to the phylogenetic analysis. The results will help to correctly identify varieties commonly used in East Africa and might be used as references to select materials and supplement data for breeding programs.

## METHODS

The dataset consisted of 33 cassava plants including Tanzanian, Kenyan and Mozambican samples. In addition, six chloroplast genomes from cassava and cassava relatives were retrieved from GenBank and added to the final dataset as references (EU117376, SRR2847471, SRR2847474, SRR2847450, SRR2847419 and SRR2847403) (*Daniell et al., 2008*; *Wang et al., 2014*). Further details about the samples used in the analysis are found in Table 1.

Field samples collection were performed differently for each batch of samples and the methods have been described in previous studies for Tanzanian (*Ndunguru et al., 2015*), Kenyan (*Ateka et al., 2017*) and Mozambican (*Amisse et al., 2019*) samples.

RNA extraction was performed from approximately 100 mg of cassava leaf as described by the cetyltrimethyl ammonium bromide protocol (*Lodhi et al., 1994*). Total RNA quality

**Table 1 Next generation sequencing data from cassava samples collected in East Africa and cassava samples retrieved from GenBank.**

| Country/sample ID | Species/cultivar name | District/region | Virus species | No. of reads obtained | No. of reads after trimming | GenBank Accession | References | Additional information |
|---|---|---|---|---|---|---|---|---|
| Tanzania 03 (Tz: Ser-6) | Rumara | N/A | CBSV | 4,988,502 | 3,936,531 | Table S1 | This study | N/A |
| Tanzania 04 (Tz: Tan 19-1) | Kibandameno | N/A | CBSV | 2,735,840 | 2,279,998 | Table S1 | This study | CMD and CBSD susceptible |
| Tanzania 05 (Tz: Tan 23) | Mkunungu | N/A | UCBSV | 4,978,272 | 4,115,933 | Table S1 | This study | N/A |
| Tanzania 06 (Tz: Tan 26) | Kibandameno | N/A | CBSV | 2,732,618 | 2,249,619 | Table S1 | This study | CMD and CBSD susceptible |
| Tanzania 07 | Unknown | N/A | Not detected | 3,500,480 | 3,031,625 | Table S1 | This study | N/A |
| Tanzania 08 | Unknown | N/A | Not detected | 5,719,724 | 4,387,729 | Table S1 | This study | N/A |
| Tanzania 09 (Tz: Nya-38) | Mkangawandu | N/A | CBSV | 2,560,544 | 2,126,102 | Table S1 | This study | N/A |
| Tanzania 10 (Tz Mf-49) | Kibembe | N/A | CBSV | 2,088,040 | 1,752,853 | Table S1 | This study | N/A |
| Tanzania 11 (Tz: Maf-51) | Kibembe | N/A | UCBSV | 2,071,164 | 1,852,435 | Table S1 | This study | N/A |
| Tanzania 12 (Tz: Maf-58) | Mwarabu | N/A | UCBSV | 2,548,594 | 2,030,993 | Table S1 | This study | N/A |
| Kenya 01 | Local | Bumula (Western) | CBSV and UCBSV | 15,573,022 | 15,282,670 | Table S1 | This study | N/A |
| Kenya 02 | Magana | Bumula (Western) | CBSV-Tz | 18,381,178 | 17,703,620 | Table S1 | This study | N/A |
| Kenya 03 | Tereka | Bumula (Western) | N/A | 17,711,474 | 17,024,115 | Table S1 | This study | N/A |
| Kenya 04 | Magana | Teso (Western) | UCBSV | 22,509,994 | 21,799,542 | Table S1 | This study | N/A |
| Kenya 05 | Magana | Busia (Western) | UCBSV and CBSV | 22,191,388 | 21,848,109 | Table S1 | This study | N/A |
| Kenya 07 | Local | Bondo (Nyanza) | EACMV-Ug | 21,488,968 | 21,240,329 | Table S1 | This study | N/A |
| Kenya 10 | Local | Bondo (Nyanza) | EACMV-Ug | 23,427,360 | 23,129,332 | Table S1 | This study | N/A |
| Kenya 11 | Kibandameno | Malindi (Coast) | EACMZV, EACMV-Ke, CBSV, UCBSV | 19,856,998 | 19,163,125 | Table S1 | This study | CMD and CBSD susceptible |
| Kenya 12 | Local | Malindi (Coast) | UCBSV and EACMV-Ke | 19,187,524 | 18,865,588 | Table S1 | This study | N/A |

| Country/sample ID | Species/cultivar name | District/region | Virus species | No. of reads obtained | No. of reads after trimming | GenBank Accession | References | Additional information |
|---|---|---|---|---|---|---|---|---|
| Kenya 13 | Kibandameno | Msambweni (Coast) | UCBSV and CBSV-Kilifi (Kenya) | 21,512,336 | 21,261,827 | Table S1 | This study | CMD and CBSD susceptible |
| Kenya 14 | Kibandameno | Msambweni (Coast) | CBSV and EACMV-Ke, UCBSV | 22,057,126 | 21,755,991 | Table S1 | This study | CMD and CBSD susceptible |
| Kenya 15 | Kibandameno | Msambweni (Coast) | CBSV | 22,124,706 | 21,851,119 | Table S1 | This study | CMD and CBSD susceptible |
| Tanzania DRJL030 | Unknown | N/A | Healthy | 7,329,128 | 6,584,936 | Table S1 | This study | Tall cassava-healthy |
| Mozambique 4 | Ezalamalithi | Namapa | CBSV | 2,460,222 | 2,406,924 | Table S1 | This study | N/A |
| Mozambique 5 | Calamidade | Namapa | CBSV | 3,608,398 | 3,562,380 | Table S1 | This study | N/A |
| Mozambique 8 | Calamidade | Nampula | CBSV | 1,541,978 | 1,519,424 | Table S1 | This study | N/A |
| Mozambique 10 | Buana | Gile District | Not detected | N/A | N/A | Table S1 | This study | N/A |
| Mozambique 11 | Fernando | Alto Molocue District | Not detected | N/A | N/A | Table S1 | This study | N/A |
| Mozambique 16 | Bwana | Alto Molocue | CBSV | 2,591,794 | 2,563,714 | Table S1 | This study | N/A |
| Mozambique 17 | Mulaleia | Mocuba District | Not detected | N/A | N/A | Table S1 | This study | N/A |
| Mozambique 20 | Cadri | Mocuba | CBSV | 2,872,090 | 2,826,912 | Table S1 | This study | N/A |
| Mozambique 23 | Mulaleia | Quelimane | CBSV | 2,635,892 | 2,619,932 | Table S1 | This study | N/A |
| Mozambique R1* | Manihot glaziovii | Namapa | CBSV | N/A | N/A | Table S1 | This study | CBSD and CMD resistant |
| Unknown_TME3 | TME3 | N/A | N/A | N/A | N/A | EU117376 | Daniell et al. (2008) | Derived from the original cassava germplasm transferred to Africa or recent African breeding improvement |
| Tanzania tree cassava | Tree cassava | N/A | N/A | N/A | N/A | SRR2847471 | Bredeson et al. (2016) | Presumed to be an M. glaziovii–M. esculenta hybrid |
| Australia TMS I50395-Unk | TMS I50395/Unk | N/A | N/A | N/A | N/A | SRR2847474 | Bredeson et al. (2016) | N/A |

(Continued)

| Country/sample ID | Species/cultivar name | District/region | Virus species | No. of reads obtained | No. of reads after trimming | GenBank Accession | References | Additional information |
|---|---|---|---|---|---|---|---|---|
| Unknown namikonga | Namikonga | N/A | N/A | N/A | N/A | SRR2847450 | Bredeson et al. (2016) | Arose by introgression of M. glaziovii into M. esculenta, derived from the Amani program. It's CBSD-resistant but CMD-susceptible |
| Tanzania M. glaziovii | M. glaziovii | N/A | N/A | N/A | N/A | SRR2847419 | Bredeson et al. (2016) | CBSD and CMD resistant |
| Brazil FLA444-1 | FLA 444-1 | N/A | N/A | N/A | N/A | SRR2847403 | Bredeson et al. (2016) | M. esculenta ssp. flabellifolia. Brazilian accession of the wild progenitor species of cassava |

check was performed followed by cDNA libraries preparation using the IlluminaTruSeq Stranded Total RNA Sample Preparation kit (Illumina, San Diego, CA, USA). Paired end reads were generated using the Illumina MiSeq System. The methods used for Next Generation Sequencing were previously described (*Ndunguru et al., 2015*; *Ateka et al., 2017*).

The raw transcriptome data was trimmed and de novo assembled using the software CLC Genomics Workbench v9.5.2, as previously described (*Ateka et al., 2017*; *Ndunguru et al., 2015*). Assembled contigs were mapped to an existent cassava chloroplast genome from GenBank (EU117376) (*Daniell et al., 2008*) cultivar TME3 and merged to produce a single draft genome using Geneious v9.1.3 (*Kearse et al., 2012*). Assemblies were refined by repeatedly mapping trimmed raw reads to the draft sequencing and adjusting as necessary. Mapping was performed with the following settings in Geneious software; minimum overlap 10%, minimum overlap identity 80%, allow gaps 10%, fine tuning set to iterate up to 10 times at custom sensitivity. A consensus between the mapped trimmed reads and the reference was used to form the new draft chloroplast genomes.

Draft chloroplast genomes were annotated using CpGAVAS (*Liu et al., 2012*). Subsequently, pairwise chloroplast alignment among the 39 cassava chloroplast genomes was performed using MAFFT v7.222 (*Katoh et al., 2002*) within the Geneious. Further annotations adjustment was performed by direct comparison with the reference genome (EU117376). Finally, the alignment was refined removing regions and genes with gaps or artifacts for further phylogeny reconstruction.

Phylogenetic Bayesian analyses were performed on DNA sequences of 55 protein-coding genes for a dataset with 39 taxa using ExaBayes version 1.4.1 (*Aberer, Kobert & Stamatakis, 2014*) and were run in parallel across 384 nodes on the Magnus supercomputer (located at the Pawsey Centre, WA, Australia). Analyses were run for 1 million generations with sampling every 500 generations. Each analysis consisted of four independent runs, each utilizing four coupled Markov chains. The run convergence was monitored by finding the plateau in the likelihood scores (standard deviation of split frequencies <0.0015). The first 25% of each run was discarded as burn-in for the estimation of a majority rule consensus topology and posterior probability for each node. Bayesian run files are available in Supplemental Files.

For virus identification on each sample, assembled contigs were sorted by length and the longest subjected to a BLAST search (blastn and blastx) (*Altschul et al., 1990*). In addition, reads were also imported into Geneious v9.1.3 (*Kearse et al., 2012*) and provided with reference sequences obtained from GenBank (NC012698 for CBSV, GQ329864 for CBSV-T and NC014791 for UCBSV). Mapping was performed with minimum overlap 10%, minimum overlap identity 80%, allow gaps 10% and fine tuning set to iterate up to 10 times.

# RESULTS

Illumina sequencing of libraries prepared from total DNA produced between 2,071,164 and 23,427,360 paired-end reads with a maximum sequence length of 100 nucleotides and minimum of 30 nucleotides for Kenyan samples, maximum sequence length of

300 nucleotides and minimum of 100 nucleotides for Tanzanian samples, maximum of 300 and minimum of 100 nucleotide sequence length for the Mozambican samples.

The phylogeny reconstruction consisted of an 35,439 bp alignment and included 55 protein-coding genes for a dataset with 39 taxa. The single-copy genes analyzed (53 genes) were *psbA*, *atpA*, *atpF*, *atpH*, *atpI*, *rps2*, *rpoC1*, *psbM*, *psbD*, *psbC*, *psbZ*, *rps14*, *psaB*, *psaA*, *ycf3*, *rps4*, *ndhJ*, *ndhK*, *ndhC*, *atpE*, *atpB*, *rbcL*, *psaI*, *ycf4*, *cemA*, *petA*, *psbJ*, *psbL*, *psbF*, *psbE*, *petG*, *psaJ*, *rpl33*, *rpl20*, *clpP*, *psbB*, *psbT*, *psbN*, *psbH*, *petB*, *rps11*, *rpl36*, *rps8*, *rpl14*, *rpl16*, *rps3*, *rpl22*, *psaC*, *ndhE*, *ndhG*, *ndhI*, *ndhA* and *ndhH*. The genes duplicated in the IR analyzed (2 genes) were *rps7* and *rpl23*. All the GenBank Accessions from the nucleotide sequences obtained in this study are available at Table S1.

Single nucleotide polymorphisms (SNP's) were evaluated by the number of differences between each of the samples, calculated by Geneious v9.1.3 (Fig. S1).

We carried out a polymorphism analysis focusing on the comparison between the cultivated cassava and the two *M. glaziovii* samples, which are cassava wild relative species. *M. glaziovii* (also named as India rubber tree species) was domesticated in South America and imported to East Africa in the early 20th Century. Since then, it has been used in cassava breeding programs to select the natural resistance of *M. glaziovii* to cassava pathogens, such as plant viruses (Nichols, 1947; Bredeson et al., 2016).

It was found a total of 31 unique SNPs that were present only in the *M. glaziovii* chloroplast coding regions and absent in all the cultivated cassava samples.

Among all the coding-regions, the *rbcL*, which encodes the large subunits of RuBisCO, was the most variable, containing four unique SNP's in the *M. glaziovii* samples. Followed by the *psbC*, *clpP*, *petB* and *psbB* with three unique SNP's each; *cemA* with two SNP's and an one nucleotide deletion; *rps3*, *ycf4*, *rpl20* with two unique SNP's each; *atpB* with one SNP and a six nucleotide deletion; and *psaB*, *psbA*, *atpA*, *atpF*, *rpoC1*, *rps4*, *psbH*, *rps11*, *rps8*, *rpl22*, *rpl23*, *ndhH* and *rpl23* with only one SNP each.

The nucleotide phylogenetic analysis clustered samples in three main clades, herein named as clades A, B and C (Fig. 1).

## DISCUSSION

We obtained the draft chloroplast genomes from 33 cultivated cassava and wild relative species belonging to three different African countries. Most of the plants analyzed were naturally infected by the two main viral diseases affecting cassava production in Africa: CBSD and CMD. We found considerable genetic diversity in the chloroplast coding regions analyzed among cultivated cassava. Additionally, no direct relationship was observed between chloroplast coding regions and virus species, indicating the need of enhancing the diversity of germplasm available.

The polymorphism analysis revealed *rbcL* as the most variable chloroplast coding-region in the wild relative species samples. The *rbcL* encodes the large subunits of RuBisCO and have been identified before to interact with the coat protein of the *Potato virus Y* (Feki et al., 2005) and it may be involved in the production of mosaic and chlorosis symptoms. In addition, *rbcL* was reported to interact with the P3 protein of other Potyviruses such as *Onion yellow dwarf virus*, *Soybean mosaic virus*, *Shallot yellow stripe*

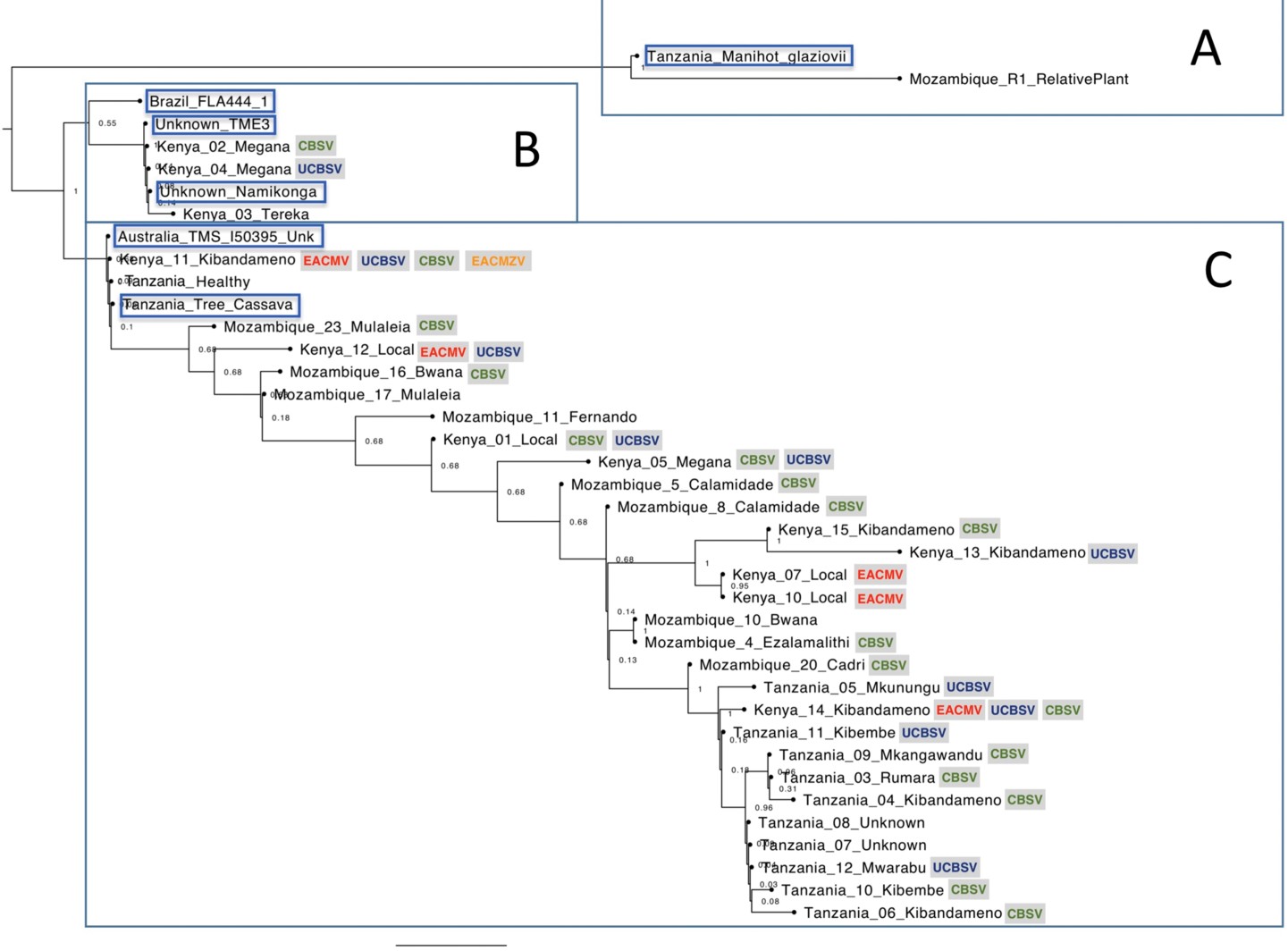

**Figure 1 Phylogenetic relationships among cassava chloroplast coding regions collected in East Africa.** The cassava plants were clustered in three main clades (A–C). The viruses detected on the samples are indicated in colorful letters inside gray squares: *Ugandan cassava brown streak virus* (UCBSV), *Cassava brown streak virus* (CBSV), *East African cassava mosaic virus* (EACMV-Ug) and *East African cassava mosaic Zanzibar virus* (EACMZV). Samples obtained from GenBank are highlighted in blue.                                   

*virus* and *Turnip mosaic virus* (*Lin et al., 2011*). It is possible that the potyvirus P3 protein affects the normal functions of RubisCO, contributing to symptom development (*Lin et al., 2011*). Curiously, Potyviruses and Ipomoviruses (causal agents of CBSD) are members of the same family, *Potyviridae*, indicating that the *rbcL* coding-region could potentially be a source of resistance or tolerance to viruses in the wild cassava *M. glaziovii*.

The clade "A" of the phylogenetic tree consisted from the wild relative species *M. glaziovii* from Tanzania and Mozambique. The clade "B" of the Phylogenetic Tree (Fig. 1) was composed by two sub-clades. The upper sub-clade containing "Brazil FLA 444-1" which is a wild progenitor species of cassava, *M. esculenta* ssp. *flabellifolia* from Brazil; and the bottom sub-clade composed by the varieties "Namikonga", "TME3" and

three cassava plants from Kenya. The SNP's analysis based on the chloroplast coding-regions revealed that "Namikonga", "Kenya 02 Megana" and "Kenya 04 Megana" are 100% identical among them. The variety "Namikonga" is known to be CBSD-resistant but CMD-susceptible and has emerged by the introgression of the wild *M. glaziovii* into *M. esculenta* (*Bredeson et al., 2016*). Curiously, both "Megana" cassava plants, which are 100% identical to Namikonga were naturally infected by CBSD, indicating that the single copy genes analyzed may not be related to the CBSD resistance found in Namikonga.

The clade "C" consisted mainly of cassava samples, excepted for "Tanzania Tree Cassava" (SRR2847471) which is supposed to be an *M. glaziovii*–*M. esculenta* hybrid (*Bredeson et al., 2016*). Other three samples ("Kenya 11 Kibandameno", "Unknown Healthy" and "Australia TMS I50395") showed 100% identity with "Tanzania Tree Cassava" on the SNP's analysis (Fig. S1). It is known that DNA barcodes derived from the chloroplast genome can be used to identify varieties (*Daniell et al., 2016*). The 100% identity among chloroplast regions from these four cassava plants indicates that they are all related to *M. glaziovii*–*M. esculenta* hybrids. Curiously, "Kenya 11 Kibandameno" was naturally infected with EACMZV, EACMV-K, UCBSV and CBSV. It suggests that this variety, which is related to the *M. glaziovii*–*M. esculenta* hybrid, is not resistant to EACMZV, EACMV-K, UCBSV and CBSV isolates present in the field in Kenya.

The variety Kibandameno seems to either present high variability or to be misidentified in Kenya and Tanzania. The cassava plant identified as "Kenya 11 Kibandameno" was placed distantly from other plants also classified as Kibandameno and showed high numbers of SNP's (46) compared with "Kenya 15 Kibandameno" (Fig. S1), suggesting that some varieties might be losing their purity. This also happened to the variety Megana in Kenya. The samples "Kenya02 Megana" and "Kenya 04 Megana" had no SNP's between them. However, 20 SNP's were present between "Kenya02 Megana" and "Kenya 05 Megana".

In general, our phylogenetic analysis suggests that chloroplast coding-regions from East-African cassava varieties present considerably genetic diversity (Fig. 1). Cassava plants from Tanzania were more genetically similar to each other compared to the other countries. Kenya is the country where cassava presented the most genetic diversity, which increases the chances of finding natural resistance associate to plant pathogens. Additionally, the data also revealed that cassava plants collected in Kenya are genetically closer to cassava plants from Mozambique. Curiously, Kenya and Mozambique are geographically more distant from Tanzania.

## CONCLUSIONS

Although a considerably diversity in the chloroplast of cultivated cassava has been found, there still a need for the introgression of new wild relative plants to increase the genetic diversity in East Africa and find new target genes resistant to CBSD and CMD. The polymorphism analysis revealed *rbcL* as the most variable chloroplast coding-region in the cassava wild relative species and could potentially be a source of resistance to pathogens. In addition, the data obtained in this study combined with more phenotypical

data, such as resistance and cultivars traits of previously characterized genotypes, may help breeding programs to achieve resistance for new cassava varieties.

### Funding
Computational resources provided by the Pawsey Supercomputing Centre with funding from the Australian Government and the Government of Western Australia supported this work. The funders had no role in study design, data collection and analysis, decision to publish, or preparation of the manuscript.

### Grant Disclosures
The following grant information was disclosed by the authors:
Australian Government and the Government of Western Australia.

### Competing Interests
Laura M. Boykin is an Academic Editor for PeerJ.

### Author Contributions
- Bruno Rossitto De Marchi conceived and designed the experiments, performed the experiments, analyzed the data, prepared figures and/or tables, authored or reviewed drafts of the paper, and approved the final draft.
- Tonny Kinene conceived and designed the experiments, performed the experiments, analyzed the data, authored or reviewed drafts of the paper, and approved the final draft.
- Renate Krause-Sakate conceived and designed the experiments, analyzed the data, authored or reviewed drafts of the paper, and approved the final draft.
- Laura M. Boykin conceived and designed the experiments, performed the experiments, analyzed the data, authored or reviewed drafts of the paper, and approved the final draft.
- Joseph Ndunguru performed the experiments, analyzed the data, authored or reviewed drafts of the paper, and approved the final draft.
- Monica Kehoe performed the experiments, analyzed the data, authored or reviewed drafts of the paper, and approved the final draft.
- Elijah Ateka analyzed the data, authored or reviewed drafts of the paper, and approved the final draft.
- Fred Tairo performed the experiments, analyzed the data, authored or reviewed drafts of the paper, and approved the final draft.
- Jamisse Amisse analyzed the data, authored or reviewed drafts of the paper, and approved the final draft.
- Peter Sseruwagi performed the experiments, analyzed the data, authored or reviewed drafts of the paper, and approved the final draft.

## Data Availability

The sequences and ExaBayes run files are available in the Supplemental Files and at GenBank: MK427095–MK427688, MK430183, MK430185–MK430413, MK455191–MK455751, MK470119–MK470547.

## Supplemental Information

Supplemental information for this article can be found online at http://dx.doi.org/10.7717/peerj.8632#supplemental-information.

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
