# Peer review of "Genetic diversity and SNP’s from the chloroplast coding regions of virus-infected cassava"

_PeerJ, doi:10.7717/peerj.8632_

## Round 0.1 · original submission · Major Revisions

Authors need to prepare a new improved version of the article taking into account the recommendations of both reviewers.

Reviewer 1 ·

Basic reporting

The report is clear, unambiguous, and well-written. The introduction provides almost comprehensive informations on the background to show the context of the research. I suggest to add in this section some informations about chloroplast genes involved in plant-virus interactions.
The literature is generally well referenced and relevant. Hovewe, ininitials of names have been left in some references (lines 56, 66, 77).
I also suggest to add some more recent literature on the subject of the study.
The figures are relevant, well labeled and described.
The gene names should be written in italics (line 185-189).

Experimental design

Research question is well defined. However, I have some comments.
In line 107 the authors said „Complete chloroplast genome sequences are useful tools for improving our understanding of the phylogenetic relationships between closely related taxa and the evolution of plant species…”. Hovewer, recent studies have successfully proven that cpDNA non-coding sequences were more effective in illuminating phylogeny of land plants than cpDNA sequences frequently used in phylogenetic studies (e.g. Xu C, Dong WP, Li WQ, Lu YZ, Xie XM, Jin XB, et al., Front. Plant Sci. 2017;8:15. 10.3389/fpls.2017.00015 pmid:28154574). Moreover, in line 241 the authors said „Previous studies reported an extraordinarily limited global chloroplast diversity of cultivated cassava (Bredeson et al. 2016)”.
Therefore, use of such research approach in my opinion is debatable and thus impossible to achieve the research goal.
How was the selection of the material for the study made? Why the numbers of samples are not equal and why only a few uninfected plants have been included in the experiment?

Validity of the findings

The Results and discussion section is actually limited to describing the results. In my opinion, information from cassava and / or cpDNA similar studies should be added. This will make the discussion more valuable.
Line 245: „… there was no direct relationship between cassava chloroplast diversity and the natural infection of a determined virus species, suggesting that the chloroplast might not work as a good molecular marker for virus-resistance in cassava.” This statement is speculative. Should be verified by analysis of other cpDNA regions.

Additional comments

In summary, I consider manuscript valuable and for future work this is perhaps a thorough starting point.

Reviewer 2 ·

Basic reporting

the work is a coherent development of the results obtained, however, they do not fully coincide with the research hypothesis.
the cited literature is well selected, the chapter "introduction" contains all necessary information and introduces the reader to the scope of the issues raised in the work.
the article structure is correct.

Experimental design

the purpose of the research is clearly defined. The authors write that "The goal of this study was to evaluate the genetic diversity of cassava chloroplast coding regions from plants naturally infected with viruses from different varieties collected in the field in East Africa." however, the context of the work shows that the results obtained should be helpful in obtaining virus-tolerant varieties. in my opinion the purpose of the work should be more specific, or the manuscript should not contain such theses.
the research material is well selected, but should be supplemented with genotypes previously characterized as resistant, they would be the basis for identifying regions specific for plant resistant or tolerant to viruses - such data could be used by breeders to select varieties.
what was the key of line selection - reference sequences? whether these forms were resistant or represented a wide range of genetic diversity?
the research methodology is well described and does not raise any objections

Validity of the findings

The research material was crop forms mostly infected with viruses. description of results and discussion focus only on the analysis of genetic similarity and its relation to the geographical distribution of the analyzed genotypes. in my opinion there is no reference to the possibility of identifying regions specific for resistant or sensitive forms and the possibility of using these results in practical breeding.
the discussion is very poor. The authors discuss the results obtained with only one publication (Bredeson et al. 2016) on almost identical topics, which excludes the innovative nature of the conducted research. perhaps the role of chloroplasts in the plant's response to a pathogen attack should be discussed in more depth, and the obtained results may refute or support the thesis.

Additional comments

the work should be rewritten. it would be more valuable to highlight differences between resistance and sensitive genotypes than focusing on a simple genetic similarity analysis

---

## Round 0.2 · accepted · Accept

The authors referred to the reviewers' comments and corrected the manuscript appropriately. The manuscript may be published in its current form.

Reviewer 1 ·

Basic reporting

The report is clear and well-written. My suggestions were taken into account. I have no comment

Experimental design

Research question is well defined. My suggestions were taken into account. I have no comment.

Validity of the findings

The Results and discussion section is now well-written. My suggestions were taken into account. I have no comment

Additional comments

In summary, I consider manuscript valuable and for future work this is perhaps a thorough starting point.

Reviewer 2 ·

Basic reporting

the authors referred to the reviewers' comments and corrected the manuscript. in my opinion the manuscript may be published in its current form

Experimental design

-

Validity of the findings

-

Additional comments

-